# Cytokines Measured in Nasal Lavage Compared to Induced Sputum in Patients with Mild Cystic Fibrosis

**DOI:** 10.3390/ijms252011081

**Published:** 2024-10-15

**Authors:** Teresa Fuchs, Artemis Vasiliadis, Manuela Zlamy, Anja Siedl, Katharina Niedermayr, Dorothea Appelt, Verena Gasser, Johannes Eder, Helmut Ellemunter

**Affiliations:** 1Department of Child and Adolescent Health, Paediatrics III, Cystic Fibrosis Centre Innsbruck, Medical University of Innsbruck, 6020 Innsbruck, Austria; 2Praxis Dr. Zlamy, Kirchstrasse 10, 6091 Götzens, Austria; 3Department of Child and Adolescent Health, Pediatrics III, Tirol Kliniken, Cystic Fibrosis Centre Innsbruck, 6020 Innsbruck, Austria

**Keywords:** cystic fibrosis, cytokines, nasal lavage, induced sputum, serum, inflammation

## Abstract

The measurement of cytokines in induced sputum and nasal lavage (NL) samples has been performed for years in people with cystic fibrosis (CF). The aim of this study was to directly compare sputum and NL samples and interpret results based on disease severity in patients who were categorized as having mild or severe lung disease. The categorization was based primarily on structural abnormalities detected on lung computed tomography and secondarily on lung function. The serum inflammatory markers neutrophil elastase (NE), IL-1β, 2, 6, 8, 10 and 17a were measured in each sputum and NL sample. Thirty-two sample pairs from 29 patients were included in this study (13 mild, 19 severe). In the patients classified as severe, many systemic inflammatory markers as well as sputum cytokines were significantly higher compared to those in the mild patients. However, all the markers measured in the NL were higher in the mild patients (*p* =< 0.05 for NE, IL-6 and IL-8). In addition, many cytokines in the NL correlated negatively with those in the sputum samples. Major differences in the cytokine levels were shown although the samples were obtained at the same time in the same patient. Advanced structural lung disease was closely related to systemic and lower airway inflammation, whereas preserved lung function was associated with higher levels in the NL. We hypothesize that the main part of the immune response takes place in the nasal mucosa in patients with minor pulmonary changes. Our results suggest that inflammation must be interpreted individually depending on the compartment in which it is measured. Further research is needed to accurately understand inflammatory markers measured in NL.

## 1. Introduction

Cystic fibrosis (CF) is a genetic disease caused by a loss of function of the CF transmembrane conductance regulator (CFTR) protein that transports chloride and bicarbonate. Hallmarks of the disease include a progressive inflammatory response and repeated infections [1]. The frequent monitoring of bronchopulmonary pathogens and correspondingly starting antibiotic therapy early has resulted in good infection control whereas the detection of inflammatory processes has proven to be more difficult [2]. CF airway inflammation starts early in life, often without clinical signs or the concurrent presence of pathogens. Cumulative evidence suggests that mucus plugs can be an important predisposing factor for inflammatory processes and may start the vicious cycle of inflammation, infection and bronchiectasis that characterizes the disease [2]. Inflammation is dominated by neutrophils, which gain a pathological phenotype, causing the exocytosis of granules, the loss of phagocytic abilities and delayed apoptosis [3]. Neutrophil elastase (NE) is one of the enzymes released excessively by granules and therefore plays an important role in CF airway inflammation [3]. NE has been shown to be one of the few markers that correlate with long-term outcomes in paediatric patients [4,5,6]. Due to the pathological function of neutrophils, they undergo necrosis rather than apoptosis, releasing intracellular substances such as interleukins (IL) [7]. A number of cytokines have been identified in the CF airways, such as IL-1β, IL-2, IL-6, IL-8 and IL-17a as the most investigated [8,9]. Much research has been carried out to analyze the relationship of inflammatory markers in people with CF and healthy controls. Specifically, IL-8, tumour necrosis factor α (TNFα) and NE were among the most frequently investigated, showing significantly higher amounts in CF patients compared to healthy controls [10,11]. Finding markers that correlate with clinical outcome parameters has been challenging as many studies provide conflicting results [8]. Correlations between biomarkers and airway pathogens showed that chronic airway *Pseudomonas aeruginosa* (Pa) load was associated with increased NE and IL-17a and *Staphylococcus aureus* was associated with increased IL-1β and vascular endothelial growth factor (VEGF) [12]. Despite successful research activity in recent years, progress has been limited not only by the complexity of the pathological CF immune system but also by the problem of finding suitable sample materials. Inflammatory markers can be measured systemically in the serum or locally by means of sputum, bronchoalveolar lavage (BAL) or nasal lavage (NL) samples. Due to highly effective modulator therapies, such as ELX/TEZ/IVA, patients with CF tend to produce less sputum. Alternative sample materials are therefore the subject of active research and NL has been used successfully in several publications [13,14,15,16]. However, there are few published studies that focus on the comparison and interpretation of markers measured in various sample materials [13,17,18]. Even less studies focus on differences according to the clinical state and disease severity.

We therefore aimed to compare systemic and local inflammatory markers measured on the same day in the same patient. The results were interpreted based on the disease severity as the patients were categorized as mild or severe.

## 2. Results

### 2.1. Study Population

Thirty-two NL and sputum sample pairs obtained on the same day were available from 29 paediatric and adult patients. An additional three sample pairs were collected three months after the first study visit. According to the selection criteria for mild and severe lung disease defined in this study, 11 patients (13 samples) were assigned to the mild and 18 (19 samples) to the severe group. Data on lung function, LCI and CT scores were obtained within the clinical routine on the same day as sample collection (FVC, FEV_1_) or within six months prior to the study visit (LCI_2.5_, CT scores). Significant differences regarding these clinical data were shown between the groups. The proportion of paediatric and adult study participants and the age distribution were balanced. However, there were more female participants in the severe group. No differences were shown regarding the occurrence of pancreatic insufficiency, CF-related diabetes, or CF liver disease. Almost all the patients with severe lung involvement displayed chronic bronchopulmonary colonization, with Pa as the most common pathogen, whereas only one patient in the mild group met the criteria for chronic colonization according to Lee et al. [19]. Therefore, a significant difference between the groups with regard to long-term antibiotic inhalation therapy was shown. Twelve patients in the severe group received modulator therapy compared to only three patients in the mild group. Only one patient with severe lung disease received azithromycin therapy and none in either group received ibuprofen therapy. The patients’ characteristics are shown in Table 1. Written informed consent was obtained from each patient and/or their legal guardian. This study was approved by the local ethics committee (EK Nr: 1055/2022).

### 2.2. Different Responses of Inflammatory Markers Depending on Sample Origin

Inflammatory markers in the serum (IgG including subtypes IL-2R, IL-6, IL-8, IL-10 and TNFα) were measured in blood samples taken as part of the clinical routine during outpatient visits. Most of the parameters measured systemically showed higher values in the patients with severe lung disease; these were statistically significant for IgG, IgG1 and IgG2 (each *p* < 0.05, Figure 1a–c). Inversely, the IL-8 levels were higher in the patients with mild disease progression, showing a weak but present significance (*p* = 0.049).

The cytokine profile measured in the sputum samples showed higher values for most of the parameters in the patients with severe lung disease. IL-1β and IL-2 (each *p* < 0.05) as well as IL-10 and IL-17a (each *p* < 0.001) showed the most significant differences (Figure 2a–d).

In the NL samples, a different picture with a tendency towards higher values of inflammatory parameters were measured in the mild group; these were statistically significant for NE, IL-6 and IL-8 (each *p* < 0.05, Figure 2e–g). The measured values are shown in the Appendix A.

### 2.3. Correlations within Markers and with Clinical Parameters

A comparison of all the correlations showed that there were hardly any similarities between the groups. Looking at the correlations between the sputum and NL samples, a large number of negative correlations were shown in both groups. Statistically significant and therefore particularly striking was the negative association of the NL IL-10 with the sputum IL-10 (r = −0.58, *p* < 0.05) and IL-17a (r = −0.67, *p* < 0.01) in the patients in the severe group. An overview of the most striking correlations within the airway and systemic cytokines is shown in Appendix A.

Correlating the systemic cytokines with the clinical parameters at the baseline in the mild patients showed a positive correlation of the total IgG and IgG1 with the CT bronchiectasis score (each r = 0.61, *p* < 0.05). High IgG3 also correlated with a worse ppFVC (r = −0.62, *p* < 0.05). No correlations were shown between the serum markers and clinical outcome parameters in the severe patients.

There were no significant correlations between the sputum markers and clinical outcome parameter in the mild patients. In the NL samples, high IL-2 correlated with worse ppFEV_1_ (r = −0.78/*p* ≤ 0.01) and high nasal IL-1β with worse ppFVC (r = −0.67/*p* ≤ 0.05).

In the patients with severe lung disease, NE measured in the sputum correlated negatively with the total CT scores (r = −0.54, *p* ≤ 0.05). Also, in the severe patients, significant correlations were shown for nasal IL-6 (with FEV_1_% and FEF 25%) and IL-8 with the FEF 25%; however, these correlations were positive (Table 2). In addition, high nasal NE values correlated with better CT scores (r = −0.54, *p* < 0.05, see Figure 3).

## 3. Discussion

This study demonstrated significant results regarding inflammatory markers measured in different body compartments in patients with different disease severity. Systemic inflammatory markers such as IgG including subclasses 1 and 2 were significantly higher in the patients with severe lung disease in our cohort (Figure 1a–d). IgG is one of the first biomarkers ever studied and high values of it were already connected to severe lung disease several decades ago [20]. More recent investigations in paediatric cohorts have confirmed that despite novel treatment standards, high IgG correlated with worse FEV_1_ [21]. These results are consistent with our findings showing that a high total IgG and IgG1 correlated with higher (worse) CT bronchiectasis scores. Also, high IgG3 correlated with a decreased ppFVC. A study by Sagel et al. showed that circulating inflammatory markers (IL-6, IL-8 and α1-antitrypsin) predicted treatment response in patients with pulmonary exacerbation [22]. This study’s findings, together with our results, show that monitoring systemic inflammatory markers should continue to play an important role in the assessment of inflammatory processes.

Due to the early diagnosis and thus the rapid start of CF specific therapy involving novel CFTR modulators, patients with CF tend to produce less sputum. Since the effect of modulators on inflammation is less clear and long-term data are lacking, the search for alternative sample material is still important and NL has been shown to be a promising predictor reflecting airway inflammation. However, the interpretation of cytokines measured in NL is still difficult and there are few studies that compare them to other sample sources. To address these issues, we compared the cytokines measured in NL and sputum samples with a focus on disease severity. We showed that local inflammatory markers measured in NL (upper airways) and induced sputum (lower airways) showed significant differences in terms of the severity of lung disease. While many cytokines measured in the sputum were significantly higher in the severe group (IL-1β, IL-2, IL-10, IL-17a; Figure 2a–d), the cytokines measured simultaneously in the upper respiratory tract were significantly higher in the mild group (NE, IL-6, IL-8; Figure 2e–g).

Initial studies in the 1990s compared cytokines in NL with BAL in paediatric patients and healthy controls and showed no differences between markers in the NL between the groups, whereas in the BAL, the markers were significantly higher in those with CF [17]. Another paediatric study addressed the same question in 2005, showing a significant relationship between IL-8 measured in NL with neutrophils and IL-8 measured in sputum, concluding that nasal IL-8 reflects lower airway inflammation [18]. The first comparisons between NL and sputum samples were published in 2014 in a mixed paediatric and adult cohort. All the markers were higher in the sputum, and significantly so for IL-8 and IL-1β. Significant positive correlations were shown between IL-8 measured in both materials. However, their cohort included patients with both mild and severe lung involvement with a mean FEV_1_ of 68% [13]. In our study, we could hardly detect any significant positive correlations between the NL and sputum samples for the same cytokines. Rather, there were many negative correlations in both groups, although not all of them were significant. A closer look at cytokines measured in NL samples revealed much higher values in patients with minor structural lung damage compared to a paediatric cohort during the stable phase in a study by Erdmann et al. [14]. The patients in this study showed a similar ppFEV_1_ but a higher rate of chronic pulmonary colonisation with Pa (23% vs. 0%) and *Staph. aureus* (53% vs. 9%). A proportion of 33.3% of the paediatric patients in this study received nebulized antibiotic treatment, while only 18.2% in our cohort received regular treatment, which may explain the significantly lower cytokine levels compared to our data.

Major limitations in our study include the small sample size, single-centre study format and the lack of healthy controls. We did not consider an unclear dilution effect of the NL as a relevant limitation, in accordance with the literature. The benefits of using the same method and thus having better comparability of the values to other studies outweighed the risk of potentially altered results. Another relevant limitation of our study is the use of DTT in sputum processing, as it is known to alter cytokine values. However, this effect has mainly been shown for TNFα [23] and, again, we preferred to use the same methodology to have better comparability to other studies. Significantly more patients assigned to the severe group received a CFTR modulator and nebulized antibiotic therapy, which both represented possible biases for the different cytokine values. Nevertheless, the systemic and sputum cytokines were significantly elevated in this group, making it difficult to interpret this potential influence.

In four patients in the severe group, NE could not be measured in the NL samples because the values were below the detection limit. We decided not to include these values in the statistical analysis, but deliberately mentioned these results to emphasize that the nasal NE was generally low or undetectable in this group.

## 4. Methods

### 4.1. Patients

Patients in this subgroup analysis are part of the INFLAM-CF study, which is conducted at the CF centre in Innsbruck, Austria. Baseline data were collected between April 2022 and October 2022. In each patient, CF diagnosis was confirmed by two positive sweat tests and/or identification of two disease-causing variants in the CFTR gene locus. Acute pulmonary exacerbation according to Fuchs criteria [24], airway infection and any kind of vaccination four weeks prior to data collection as well as systemic or inhaled corticosteroids, immunosuppressive therapy, chronic colonisation with *non-tuberculous mycobacteria* and chronic rhinosinusitis were defined as exclusion criteria. Study visits were carried out as part of the routine outpatient visits. Blood sampling, lung function tests, multiple breath washout (MBW) measurements and multi-detector computed tomography (MD-CT) scans were performed within the clinical routine.

Patients were divided into two groups. One group, defined as mild, included patients with minor structural lung damage and preserved lung function. The other group included patients with severe lung disease. The following parameters were used for classification: chest CT scores, forced expiratory volume in one second (ppFEV_1_), forced vital capacity (ppFVC) and lung clearance index (LCI_2.5_). The threshold values for assignment to the severe group were as follows: lung CT scores < 20 points (fewer points represent more pronounced lung disease), according to Bhalla et. al. [25], as well as FEV_1_ and FVC < 80% and LCI_2.5_ ≥ 8.

### 4.2. Nasal Lavage Sampling

NL sampling was carried out at the beginning of respiratory therapy by trained physiotherapists. We refer to our previous publication for more detailed information on the implementation of NL [26]. In short, 10 mL of 0.9% sodium chloride per nostril was used. Aliquots containing an additional 15 µL of protease inhibitor (Protease Inhibitor Mix G, SERVA^®^, Heidelberg, Germany) were immediately frozen at –80° C until further analysis. The performance of NL is part of daily routine therapy in patients treated at the CF centre in Innsbruck and is recommended from the point of diagnosis.

### 4.3. Sputum Processing

Induced sputum samples were obtained by nebulisation of hypertonic saline during respiratory physiotherapy following NL sampling. The mouth was thoroughly rinsed with water before starting the inhalation. Further processing of the sputum was carried out within 2 h of collection. Sputum plugs were removed from the sample and further processed with 4× wt/vol of 0.1% Dithiolthreitol (DTT, Fisher Scientific (Austria) GmbH, Vienna, Austria) and subsequently with 2× wt/vol of phosphate-buffered saline (Fisher Scientific (Austria) GmbH, Vienna, Austria). The sample was filtered through a 70 µg filter and then centrifuged at 2000 rpm for 10 min. Supernatants were frozen at –80 °C until further analysis [27].

### 4.4. Measurement of Inflammatory Markers

The cytokine profile was measured in NL and sputum samples from the same patient obtained on the same day. IL-1β, IL-2, IL-6, IL-8, IL-10, IL-17a and VEGF values were determined using Milliplex MAP-Kits ^®^ (Human Cytokine/Chemokine/Growth Factor Panel A, Merck Millipore, Darmstadt, Germany). As specified by the manufacturer, the minimum detection limits were 0.52 pg/mL (IL-1β), 0.28 pg/mL, (IL-2), 0.14 pg/mL (IL-6), 0.52 pg/mL (IL-8), 0.91 pg/mL (IL-10) and 0.71 pg/mL (IL-17A). NE was measured using PMN Elastase ELISA (DEH331, demeditec Diagnostics GmbH; Kiel, Germany) and analyzed with a FLUOstar Galaxy spectrometer (BMG LABTECH GmbH; Offenburg, Germany). The detection limit was set at 0.2 ng/mL. All tests were performed according to the manufacturer’s instructions and samples were run in duplicate. NL samples were analyzed directly without further dilution.

### 4.5. Statistical Analysis

Statistical analysis was performed with SPSS, version 29.0 (IBM, Ehningen, Germany) and Prism, version 9.5.1 (GraphPad Software Inc. Boston, MA, USA). Due to the small sample size, groups were compared using the Mann–Whitney U test and correlations were performed using the Spearman non-parametric correlation. *p* < 0.05 was considered significant.

## 5. Conclusions

In this study, we demonstrate that patients with supposedly mild CF lung disease have significantly higher cytokine levels in their upper airways compared to patients with progressed structural lung damage. Interpreting these data in view of preserved lung function with a median FEV_1_ of 99%, a median LCI of 6.5 and minimal changes in lung CT scans, we hypothesize that the main part of the immune response takes place in the nasal mucosa in patients with minor pulmonary changes. Long-term observational studies will show the extent to which this immune response affects clinical outcomes. Despite the mild course of the disease in this group, elevated cytokines should not be hastily interpreted as less relevant since high nasal IL-2 and IL-1β levels were correlated with worse lung function. With our results, we would prefer to emphasize that the nose is the gateway to the respiratory tract and an early diagnosis of inflammation in young patients is still crucial. In the patients with severe lung disease, higher nasal IL-6 and IL-8 levels were correlated with better lung function and high NE was correlated with better CT scores. In this case, the main inflammatory processes happen in the lower airways, posing the question of whether NL is a suitable sample material to monitor inflammation in this patient collective.

In conclusion, this study is one of few that compare inflammatory parameters measured in three compartments (systemic and upper and lower respiratory tracts). We claim that inflammation should be interpreted individually and analyzed according to the disease severity, focusing on different compartments. Further research is needed to accurately understand the inflammatory markers measured in NL.

## Figures and Tables

**Figure 1 ijms-25-11081-f001:**
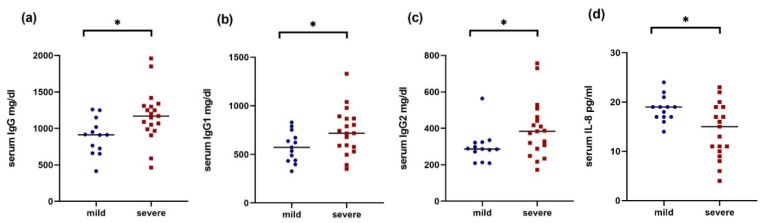
Serum IgG (**a**), IgG1 (**b**), IgG2 (**c**) and IL-8 (**d**) ^#^ in mild (blue circles) and severe (red squares) CF patients. Line represents median and *p*-values represent results of Mann–Whitney U test. Note: ^#^ 1 high outlier value has been removed in the severe group (48 pg/mL). * *p* < 0.05.

**Figure 2 ijms-25-11081-f002:**
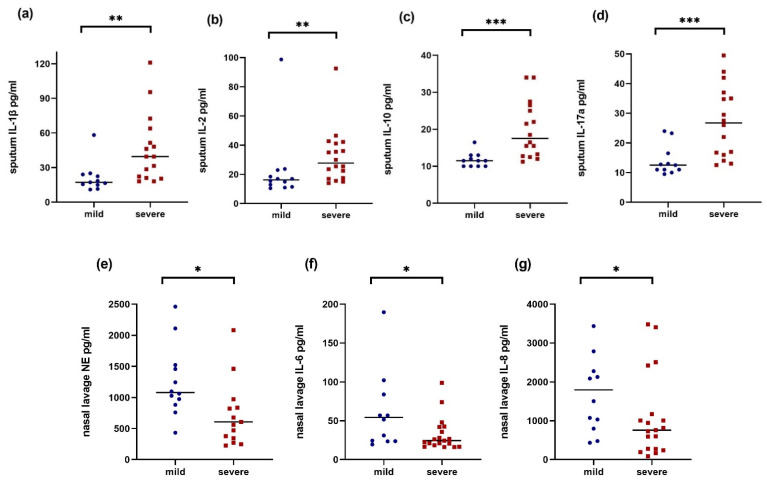
Sputum IL-1β (**a**) ^#^, sputum IL-2 (**b**), sputum IL-10 (**c**) ^#^, sputum IL-17a (**d**), NL NE (**e**) ^†^, NL IL-6 (f) ^†^, and NL IL-8 (g) ^†^ in mild (blue circles) and severe (red squares) CF patients. Line represents median and *p*-values represent results of Mann–Whitney U test. Note: NL NE: Four severe patients were excluded because the values were below the detection limit, and one high value is not displayed (3081 pg/mL). ^#^ n = 11 in mild and n = 15 in severe due to technical issues. ^†^ n = 12 in mild. * *p* < 0.05, ** *p* < 0.01, *** *p* < 0.001.

**Figure 3 ijms-25-11081-f003:**
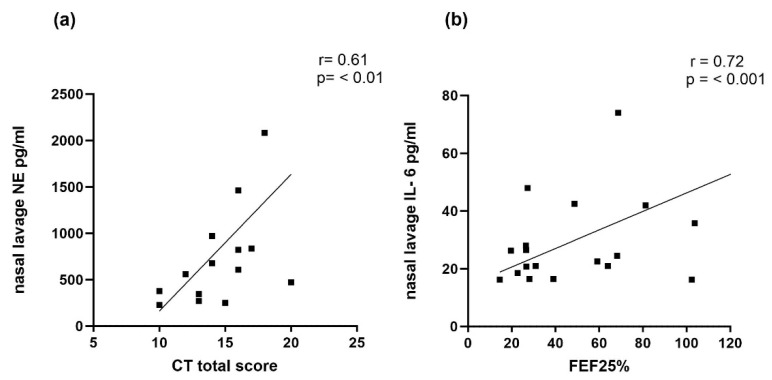
Spearman’s correlation coefficient of NL NE with CT total score (**a**) and NL IL-6 with ppFEF25 (**b**) in patients assigned to severe group. *Note:* 1 high outlier value has been removed each in ((**a**) [NE 3081 pg/mL]) and ((**b**) [FEF 25% 220.2%]).

**Table 1 ijms-25-11081-t001:** Patient characteristics and differences between the two study groups at baseline. Results are expressed as median (range) with statistical analysis calculated using Mann–Whitney U test.

Variable	Mild	Severe	*p*
n	11	18	
Sex, n (%) Male Female	8 (72.7) 3 (27.3)	4 (22.2) 14 (77.8)	<0.05
Age at inclusion, y Median (range)	17 (10–40)	25 (7–44)	>0.05
Paediatric, n (%) Adult, n (%)	6 (54.5) 5 (45.5)	5 (27.8) 13 (72.2)	>0.05 >0.05
Clinical parameters CT score, median (range)	22 (20–24)	14 (10–19)	<0.001
LCI median (range)	6.5 (5.5–8.0)	9.7 (5.5–14.4)	<0.001
FVC pred, % median (range)	100.1 (87.6–118.7)	89.6 (70.3–114.3)	<0.05
FEV_1_ pred, % median (range)	99.0 (83.6–112.5)	78.6 (48.6–121.5)	<0.05
FEF75 pred, % median (range)	91.7 (42.1–122)	35.0 (14.5–220.2)	<0.01
Chr. bronchopulmonary colonization, n (%) PSA, n (%) SA, n (%) Stenotrophomonas, n (%) Aspergillus fum., n (%) Burkholderia, n (%) Achromobacter, n (%)	1 (9) 0 (0) 1 (100) 0 (0) 0 (0) 0 (0) 0 (0)	17 (94.5) 7 (41.2) 4 (23.4) 2 (11.8) 2 (11.8) 1 (5.9) 1 (5.9)	<0.001
Inhaled antibiotics, n (%)	2 (18.2)	12 (66.7)	<0.05
Medical records PI, n (%)	9 (81.8)	17 (94.4)	>0.05
CFRD, n (%)	0 (0)	2 (11.1)	>0.05
CFLD, n (%)	4 (36.4)	5 (27.8)	>0.05
CFTR genotype dF508 homozygous dF508 heterozygous other	3 (27.3) 6 (54.5) 2 (18.2)	8 (44.5) 9 (50.0) 1 (5.5)	
Modulator therapy, n (%) Kaftrio, n (%) Kalydeco, n (%) Orkambi, n (%)	3 (27.3) 1 (9) 1 (9) 1 (9)	12 (66.7) 11 (91.7) 0 (0) 1 (8.3)	<0.05

**Table 2 ijms-25-11081-t002:** Most striking correlations of inflammatory markers with lung function, LCI and CT scores.

Variable	ppFVC	ppFEV1	ppFEF25	LCI	CT total score	CT Bronchiectasis Score
IgG	-	-	-	-	-	0.61 *
IgG1	-	-	-	-	-	0.62 *
IgG3	−0.62 *	-	-	-	-	-
SP NE	-	-	-	-	−0.54 *	-
NL NE	-	-	-	-	0.54 *	-
NL IL-1β	−0.67 *	-	-	-	-	-
NL IL-2	-	−0.78 *	−0.70 *	-	-	-
NL IL-6	-	0.51 *	0.72 ***	-	-	-
NL IL-8	-	-	0.53 *	-	−0.54 *	-

*Note:* Significant Spearman’s correlation coefficients are in red for severe and blue for mild. * *p* < 0.05, *** *p* < 0.001.

## Data Availability

The data presented in this study are available on request from the corresponding author (Please specify the reason for restriction, e.g., the data are not publicly available due to privacy or ethical restrictions).

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
