# Peer review of "Cytokines Measured in Nasal Lavage Compared to Induced Sputum in Patients with Mild Cystic Fibrosis"

_ijms, 2024, doi:10.3390/ijms252011081_

Round 1
Reviewer 1 Report
Comments and Suggestions for Authors
The research manuscript “CYTOKINES MEASURED IN NASAL LAVAGE COMPARED TO INDUCED SPUTUM IN PATIENTS WITH MILD CYSTIC FIBROSIS” describes the discrepancy in inflammatory parameters between nasal lavages, induced sputum and lung damage measured by CT and lung function. The rationale would be to validate methods to determine cystic fibrosis lung disease in other ways than with CT and avoid the drawbacks of x-ray exposure. The results were unexpected in that all markers measured in nasal lavage were higher in patients with mild lung disease than in patients with severe lung disease. There were also parameters in nasal lavage that correlated negatively with sputum samples, indicating a discrepancy between the nose and lower airways. Another interesting finding is that no correlations were found between serum markers and clinical outcome parameters in severe patients. This indicates that serum is not ideal for biomarker search. The manuscript demonstrates an interesting discrepancy and results warranting discussion of interest to the community. The references are carefully chosen and the statistical analysis correctly performed. However, there are some concerns which need to be addressed:
Major concerns
1: Figure 2: A: One high value is not displayed. Please give the value as a number, if you do not want to display it in the figure, for clarity.
B: Values below the detection limit are tricky. Should they be removed? Should they be set to half the detection limit? They should probably not be set to zero, however. This point should be further discussed and motivated by the authors.
C: Do I understand correctly that you included the high value in the analysis but removed the sample below the detection limit?
2: The groups would have to be carefully defined, as the results are not the expected, see point 3.
3: The authors state that “we hypothesize, that the main part of the immune response takes place in the nasal mucosa in patients with minor pulmonary changes”.
This can of course be hypothesized, but it was not tested in this manuscript. It would be interesting to investigate, but this manuscript demonstrates unexpected correlations between inflammation and mild disease, which need to be explained with experiments. As a consequence, the patient groups need better classification. Exactly what parameters were used to define mild and severe disease, respectively? The included patients seem to have rather mild cystic fibrosis compared to other cystic fibrosis patients, with no cystic fibrosis related diabetes or liver disease, for example. Please comment on how this cohort relates to the cystic fibrosis community in general.
Minor concerns
1: Results section: What does the acronym LCI mean? Please explain all acronyms or alternatively, refrain from using acronyms.
Author Response
Comment 1: Figure 2: A: One high value is not displayed. Please give the value as a number, if you do not want to display it in the figure, for clarity.
Response 1: thank you for the hint. I added the value as a number in the note section below the figure. I did the same in Figure 1 for IL8 for more clarity. In favor of an easily legible graphic, we decided to exclude this value.
Comment 2: Values below the detection limit are tricky. Should they be removed? Should they be set to half the detection limit? They should probably not be set to zero, however. This point should be further discussed and motivated by the authors.
Response 2: You are very right; this is always a tricky question. Since we could not detect NE in some of the samples, we concluded that despite the sensitive measurement method, this value cannot be detected and is therefore probably not present in any relevant quantity. However of course, this could be a bold assumption since we are definitely missing a control group to address this issue. We did not set them to zero, but we also did not include them in the statistical analysis since quality criteria of the analysis are not met. A paragraph addressing this was added to the end of the discussion.
Comment 3: Do I understand correctly that you included the high value in the analysis but removed the sample below the detection limit?
Response 3: Yes, because even though some of the values were higher compared to others, they were still within the detection range of the analysis kits we used. We discussed the normal distribution and ranges of these values and decided to include them in the analysis, as excluding them would not have significantly changed the overall results. Higher values still met the quality criteria of the analysis kits / methodology used, while extremely low values did not. Therefore, higher values were included in the analysis.
Two Higher values were only excluded for better presentation in the graphs.
Comment 4: The groups would have to be carefully defined, as the results are not the expected, see point 5.
Response 4: Thank you for the hint. We rephrased one paragraphs in the Methods part (“2.1 Patients”) to characterize the two groups more clearly.
Comment 5: The authors state that “we hypothesize, that the main part of the immune response takes place in the nasal mucosa in patients with minor pulmonary changes”. This can of course be hypothesized, but it was not tested in this manuscript. It would be interesting to investigate, but this manuscript demonstrates unexpected correlations between inflammation and mild disease, which need to be explained with experiments. As a consequence, the patient groups need better classification. Exactly what parameters were used to define mild and severe disease, respectively? The included patients seem to have rather mild cystic fibrosis compared to other cystic fibrosis patients, with no cystic fibrosis related diabetes or liver disease, for example. Please comment on how this cohort relates to the cystic fibrosis community in general.
Response 5: Yes, it is true that we only show the results and express hypothesis but we cannot prove it. However, as these results were also surprising for us, we are planning to use this data to carry out further research including experiments and to confirm our current results with long-term data. As we stated in the comment above, we tried to classify the two groups more clearly and rephrased the paragraph in the methods.
Lung CT score (according to Bhalla et al scoring system – DOI: 10.1148/radiology.179.3.2027992), spirometry data (FVC% and FEV1%) as well as lung clearance index (LCI) are commonly used tools to monitor lung disease in CF and it was precisely these tools that were used for classification.
When we compare median values of our patients to cohorts in other studies with similar research background, median FEV1% for example shows similar results (99.0% in mild/78.6% in severe compared to 93.4% in a CF paediatric cohort and 63.0% in an adult cohort– DOI: 10.3389/fimmu.2022.947359 supplement data). Comparison to the ECFS patient registry of 2022 showed mean FEV1 of 93.5% in paediatric patients and 70-82% in adult patients <50 years
(reference: ECFSPR Annual Report 2022, Zolin A, Adamoli A, Bakkeheim E, van Rens J et al, 2024).
Overall, we would say that we are doing our best to ensure that our cohort is comparable to the CF community in terms of clinical parameters.
Comment 6: Results section: What does the acronym LCI mean? Please explain all acronyms or alternatively, refrain from using acronyms.
Response 6: LCI means Lung Clearance Index and is first abbreviated in the section 2.1.
Reviewer 2 Report
Comments and Suggestions for Authors
A clinical study compared the inflammatory markers on nasal lavage and sputum in CF patients with normal subjects. The sample size of the study is small which may limit the reliability of the conclusion, a larger-scale study is recommended.
Some experiments related to the molecular mechanisms of the inflammatory process on the inflammatory cytokines/ IgG concentration changes in CF patients are recommended, if not feasible, are there any related published studies? If yes, the authors may mention it in the discussion.
In section 4.4, the detection limit of PMN Elastase is 0.2 mg/mL, which is extraordinarily high, is it true?
Author Response
Comment 1: A clinical study compared the inflammatory markers on nasal lavage and sputum in CF patients with normal subjects. The sample size of the study is small which may limit the reliability of the conclusion, a larger-scale study is recommended.
Response 1: Thank you for your response and feedback. In fact, unfortunately, we did not compare CF patients with healthy controls; we compared patients with mild and severe lung disease and all had the diagnosis of CF. We listed the lack of a control group as a major limitation.
Yes, you are right, the sample size is small which is also mentioned as a major limitation. A larger cohort would be preferable but this needs to be a multi-centre study since CF is still a rare disease. We will definitely work on this limitation for future studies.
Comment 2: Some experiments related to the molecular mechanisms of the inflammatory process on the inflammatory cytokines/ IgG concentration changes in CF patients are recommended, if not feasible, are there any related published studies? If yes, the authors may mention it in the discussion.
Response 2: Thank you for your comment. Unfortunately, we do not entirely understand the question or recommendation. Do you mean if there are any mechanism that influence/change cytokines concentration or if there are recommendations how to influence those mechanisms?
Of course, we would be very happy to address this concern if you could rephrase and re-send us your concern.
Comment 3: In section 4.4, the detection limit of PMN Elastase is 0.2 mg/mL, which is extraordinarily high, is it true?
Response 3: Thank you very much for drawing our attention to this error. Of course, it is 0.2 ng/ml and not mg/ml. We changed it in the main text.